# Innovative Partnerships for the Elimination of Human African Trypanosomiasis and the Development of Fexinidazole

**DOI:** 10.3390/tropicalmed5010017

**Published:** 2020-01-27

**Authors:** Philippe Neau, Heinz Hänel, Valérie Lameyre, Nathalie Strub-Wourgaft, Luc Kuykens

**Affiliations:** 1Sanofi France, 82 avenue Raspail, 94250 Gentilly, France; valerie.lameyre@sanofi.com; 2Sanofi Deutschland, Industriepark Höchst, Bldg. H831, 65926 Frankfurt am Main, Germany; Heinz.Haenel@sanofi.com; 3Drugs for Neglected Diseases initiative (DNDi), 15 Chemin Louis-Dunant, 1202 Geneva, Switzerland; nstrub@dndi.org; 4Sanofi US, 55 Corporate Drive, Bridgewater, NJ 08807, USA; Luc.Kuykens@sanofi.com

**Keywords:** sleeping sickness, human African trypanosomiasis, *T. b. gambiense*, g-HAT, *T. b. rhodesiense*, r-HAT, elimination, neglected tropical diseases, fexinidazole

## Abstract

Human African Trypanosomiasis (HAT or sleeping sickness) is a life-threatening neglected tropical disease that is endemic in 36 sub-Saharan African countries. Until recently, treatment options were limited and hampered by unsatisfactory efficacy, toxicity, and long and cumbersome administration regimens, compounded by infrastructure inadequacies in the remote rural regions worst affected by the disease. Increased funding and awareness of HAT over the past two decades has led to a steady decline in reported cases (<1000 in 2018). Recent drug development strategies have resulted in development of the first all-oral treatment for HAT, fexinidazole. Fexinidazole received European Medicines Agency positive scientific opinion in 2018 and is now incorporated into the WHO interim guidelines as one of the first-line treatments for HAT, allowing lumbar puncture to become non-systematic. Here, we highlight the role of global collaborations in the effort to control HAT and develop new treatments. The long-standing collaboration between the WHO, Sanofi and the Drugs for Neglected Diseases *initiative* (Geneva, Switzerland) was instrumental for achieving the control and treatment development goals in HAT, whilst at the same time ensuring that efforts were led by national authorities and control programs to leave a legacy of highly trained healthcare workers and improved research and health infrastructure.

## 1. Introduction

Human African Trypanosomiasis (HAT), or sleeping sickness, is a life-threatening, neglected tropical disease (NTD). It is considered endemic in 36 sub-Saharan African countries, where around 60 million people are estimated to be at some level of risk of HAT, primarily those living in ‘foci’ in poor and remote rural areas where health infrastructure is poor or non-existent [1,2,3]. HAT is transmitted to humans by the bite of tsetse flies and the etiological agent of HAT is the kinetoplastid protozoan parasite *Trypanosoma brucei (T. b.)*. Two subspecies of this parasite are pathogenic for humans: *T. b. gambiense* (g-HAT) responsible for the chronic form of the disease occurring in western and central Africa, and *T. b. rhodesiense* (r-HAT) responsible for a more acute form occurring in eastern and southern Africa [4]. *T. b. gambiense* is currently responsible for 98% of HAT cases [5], with the highest disease burden in the Democratic Republic of Congo (DRC) where 37.5 million people were estimated to be at some level of risk of g-HAT between 2012 and 2016 [1,2]. Humans are the principal reservoir for g-HAT and the progressive disease course occurs in two stages: a hemolymphatic phase (stage 1 with a mean duration of around 18 months) with common signs and symptoms including fever, headache, pruritus, weakness, asthenia, anemia, and lymphadenopathy; and a meningo-encephalic stage (stage 2 with a mean duration of around 8 months) occurring when the parasites have crossed the blood–brain barrier with resulting sleep disturbances and neuropsychiatric symptoms that may lead to coma and death if left untreated [4,5,6].

In this review, we highlight the role of global collaborations in the effort to control HAT and develop new treatments. In particular, we emphasize the role played by the public–private partnership between Sanofi (formerly Aventis) and the Drugs for Neglected Diseases *initiative* (DND*i*, Geneva, Switzerland), together with a range of governmental national sleeping sickness control programs (NSSCPs; such as the Le Programme National de Lutte contre la Trypanosomiase Humaine Africaine of the DRC [PNLTHA]) and non-governmental organizations (NGOs), in development of the first oral-only treatment for HAT, Fexinidazole Winthrop (fexinidazole).

## 2. A Strategy to Eliminate HAT

### 2.1. Global Alliances and Innovative Partnerships

Devastating epidemics of HAT have occurred throughout the 20th century. Following the neglect of control efforts implemented up to the 1960s, a dramatic resurgence of the disease was observed in the 1990s, with the World Health Organization (WHO, Geneva, Switzerland) estimating that 300,000 new cases were occurring each year in endemic areas, although only 30,000 cases were being diagnosed and treated [7]. In response to this relapse, the WHO passed a resolution in 1997 to raise awareness of the disease, promote access to diagnosis and treatment, and strengthen control and surveillance [8]. Around 20 years ago, the WHO launched an intensified coordination initiative [9] to achieve these goals by forming global alliances with United Nations Agencies (under the Program Against African Trypanosomiasis, PAAT), national governments (under NSSCPs), and the Organization of African Unity who established the Pan African Tsetse and Trypanosomiasis Eradication Campaign (PATTEC) in June 2000. Another key alliance with the WHO was the formation of what would become a longstanding public–private partnership with Sanofi in 2001, which made it possible to distribute vital drugs for the treatment of HAT and other NTDs in endemic areas free of charge. This partnership between the WHO and Sanofi also provided funds to support screening and surveillance efforts, improving treatment centers and training of local health workers, and, to contribute to research and development programs for new treatments. Under this agreement, renewed in 2006, 2011, and again 2016, Sanofi provided on average 5 million dollars per year for combating HAT (25% donated through drug donation and 75% as financial support for capacity building, patient screening, and other projects), totaling more than 85 million dollars [10]. In 2012, Sanofi cemented its commitment to the goal of eliminating HAT by signing the ‘London Declaration on Neglected Tropical Diseases’ [11]. This declaration focused on the commitment of a range of actors from public and private sectors to control or eliminate 10 infections, including HAT, affecting the world’s poorest populations. In this context, Sanofi was contributing to the elimination of HAT as a public health issue by aligning its objectives with the elimination goals of the WHO, by continuing to freely supply drugs to ensure that all patients had access to appropriate treatment, and by developing new therapeutic options, through an innovative partnership with the DND*i* for the development of fexinidazole, improving the management of the disease.

In addition to the pivotal partnership between the WHO and Sanofi, similar private collaborations were formed with Bristol-Myers-Squibb and Bayer Healthcare for the supply of raw materials and other treatments. The WHO also coordinated partnerships between NGOs and research institutions, leading to collaborations with the Bill and Melinda Gates Foundation, Doctors Without Borders (Médecins Sans Frontières [MSF], Geneva, Switzerland) and the DND*i*.

### 2.2. The Success of Collaborative Strategies for HAT Elimination

As a result of the substantial collaborative efforts of the global alliance initiative, with the key role in strengthening control and surveillance activities being played by the NSSCPs of endemic countries, there has been a steady decline in HAT cases over the past two decades (Figure 1) [2,12,13]. Since 2001, more than 40 million HAT screening tests have been performed, and over 210,000 cases detected and treated [14]. A 96% decline in reported HAT cases was observed between 2000 and 2018, with 26,550 cases reported in 2000 and a historic low of 977 cases reported in 2018 [2,15]. In 2012, HAT was included in the WHO’s roadmap for the control, elimination or eradication of NTDs and a target date was set for global elimination of g-HAT as a public health problem (<1 case/10,000 inhabitants in at least 90% of endemic foci) by 2020, with complete interruption of transmission in Africa targeted for 2030 [16]. Representatives of partners in the HAT global alliance attending the third WHO meeting of stakeholders on the elimination of g-HAT in Geneva in April 2018 concluded that global targets for reducing the number of cases had been met and that the goal of eliminating HAT was in sight [17].

### 2.3. The Need for New Treatment Options

Despite the overwhelming success of HAT initiatives in reducing the number of cases, new diagnostic tools and treatments were deemed essential to facilitate the integration of the specialized HAT control, surveillance and management programs into national health systems and to maintain the goal of g-HAT elimination [17,18]. Historically, the diagnostic tools and algorithms—and the treatments available to manage HAT—have been complex and cumbersome, particularly given that the most at-risk populations are primarily located in the most remote, resource-poor, and politically unstable regions of endemic countries [19]. Indeed, even in 2016 around 28% of the population at highest risk of g-HAT (more than 200,000 people) were >5 h travel from a competent fixed diagnostic and treatment facility [2].

One of the major drawbacks of the treatments available before the development of fexinidazole (summarized in Table 1) was that a complicated sequence of events was required before hospital treatment could be accessed: serological screening, generally performed via active surveillance, followed by laboratory-based microscopy analyses to confirm the presence of the parasite in the lymph and blood, and, finally, disease staging involving patients undergoing lumbar puncture to determine the presence of trypanosomes and the levels of white blood cells (WBCs) in the cerebrospinal fluid (CSF) [16]. In a first effort to improve treatment, which at the time consisted of melarsoprol (a highly efficient but toxic arsenic-based treatment), a short-term research and development strategy evaluating the effectiveness of combining available treatments was implemented. This initiative—instigated by MSF and Epicentre and then the DND*i*, with Sanofi and Bayer and other alliance partners—led to the nifurtimox–eflornithine combination therapy (NECT) becoming available to treat stage 2 g-HAT in 2009 [20]. Clinical trials demonstrated that NECT was a safe and effective treatment [21] and led to this combined therapy being used as a first-line therapy for stage 2 HAT in all endemic African countries. NECT helped to drastically reduce the number of HAT cases, with a steep decline in cases from around 7000 cases in 2010 to under 1000 cases in 2018 (Figure 1). However, NECT posed logistical concerns in remote locations (weight per treatment: 9 kg; volume per treatment 37.5 dm^3^; Figure 2a,b), and required hospitalization and trained nursing staff to administer [22,23].

Thus, the HAT Platform, supported by DND*i*, defined new target product profile characteristics for HAT treatments, namely for a safe treatment that was effective in both stages of HAT with the same dosing regimen and could avoid the need for systematic lumbar puncture for staging [25].

## 3. The Development of a New Oral Treatment for HAT: The Story of Fexinidazole

Fexinidazole is the first drug to be developed to fulfil this target product profile. This 5-nitroimidazole derivative DNA synthesis inhibitor [26] is the only, all-oral treatment for HAT and is indicated for use by adults and children (aged ≥ 6 years and weighing ≥ 20 kg) for both g-HAT stages [24,27]. The weight-based dose of fexinidazole (adults > 35 kg:1800 mg/d loading dose for 4 days and 1200 mg/d maintenance dose for 6 days; children aged ≥ 6 years and weighing ≥ 20 kg and <35 kg:1200 mg/d loading dose for 4 days and 600 mg/d maintenance dose for 6 days) is administered in the form of tablets to be taken once daily with food, with the treatment course being provided in the form of a simple wallet (Figure 2c–f) [24]. This revolutionary new treatment for HAT has the potential to transform disease management for both patients and healthcare workers [18,27,28]. As fexinidazole is effective for both disease stages, depending on the severity of the clinical signs, systematic lumbar puncture is no longer required for determination of the treatment strategy. Adherent patients, with adequate food intake, no psychiatric disorders or history of psychiatric disorders, or signs of advanced disease, can benefit from outpatient treatment administration under daily supervision by trained staff, reducing the risk of the disease having a major impact on their livelihood and daily activities, as well as those of their families. This simplified treatment regimen also has major benefits for healthcare infrastructure: fexinidazole is easy to store (<30 °C) and transport, and outpatient treatment will free-up vital resources for other health concerns [24,28].

### 3.1. The Rediscovery of Fexinidazole

Fexinidazole was developed over 15 years (illustrated in Figure 3), through partnership between Sanofi and the DND*i*, with clinical studies being conducted by investigators from endemic African nations coordinated by the NSSCP in the DRC, via support from the HAT Platform [29] and partners of the WHO global alliance initiative [9]. Shortly after its creation in 2003, the DND*i* began the search for a new oral treatment for HAT. The DND*i*’s expert group first identified nitroimidazoles, a group of compounds with known antiprotozoal activity, as a class to target in the search for a new HAT treatment. Among the reports identified was a review published by Raether and Hänel [30] on nitroheterocyclic compounds. One of the authors of the review (Prof. H. Hänel*,* Sanofi, Paris, France), working together with a representative of the DND*i* (Dr B. Bourdin Trunz*)*, selected fexinidazole (designated HOE239) as a potential candidate for further development [31,32]. Prof. Hänel had conducted research on fexinidazole in the 1970s as a student working for Hoechst AG Frankfurt/Main in Germany (now part of Sanofi). It was one of several hundred compounds produced by Hoechst chemists between 1950 and 1980 as part of a project to identify antiprotozoal agents [32]. The project was abandoned in the 1980s due to changes in company strategy, but not before pre-clinical tests of fexinidazole showed that this agent displayed promising efficacy against *T. brucei* infections in mice, accompanied by good oral availability and toxicity profiles [33,34]. After an extensive search of the Hoechst archives, Prof. Hänel was able to provide detailed chemical synthesis, preclinical safety, efficacy and pharmacokinetic data for Fexinidazole*,* together with a few milligrams of the compound [32].

Fexinidazole was just one of over 700 nitroheterocyclic compounds evaluated and profiled as part of the mining exercise conducted in collaboration with the Swiss Tropical and Public Health institute (Swiss TPH, Basel, Switzerland). The positive pre-clinical profile obtained for fexinidazole [35] led, in 2009, to a new agreement between Sanofi and the DND*i* for the development of fexinidazole. Under the terms of this agreement, the DND*i* was responsible for preclinical, clinical and pharmaceutical development and Sanofi was responsible for the industrialization, production, registration and distribution of the drug [27,36].

### 3.2. Clinical Trials

The first in-human clinical trials involving fexinidazole began in 2009 [37]. These phase I studies were conducted in France in healthy males of African origin to assess the safety, tolerance and pharmacokinetic properties of fexinidazole, as well as bioavailability under different food intake conditions [27,37]. After consultation on the design of the clinical development plan with both the European Medicines Agency (EMA, Amsterdam, Netherlands)–under the article 58 procedure–and U.S. Food and Drug Administration (FDA, Silver Spring, Maryland,MD, USA), in the presence of WHO observers [38], the phase II/III clinical trials of fexinidazole began in 2012 (summarized in Table 2). The efficacy of fexinidazole in the treatment of g-HAT was demonstrated by a pivotal multicenter, randomized, open-label, phase II/III trial comparing fexinidazole and NECT in adult patients (aged ≥ 15 years) with late stage 2 g-HAT (DNDiFEX004; [18,39]). As its primary efficacy outcome, based on a predetermined acceptability margin for the difference in success rates between the two treatments of −13%, this trial showed that fexinidazole met the non-inferiority objective concerning treatment success rates at 18 months. After the end of treatment, success rates of >90% were observed for both agents (91.2% for fexinidazole and 97.6% for NECT). Two non-comparative, prospective, “plug-in” trials then reported treatment success rates of 98.7% 12 months after the end of treatment in a further cohort of stage 1 or early stage 2 adult (≥15 years) g-HAT patients (DNDiFEX005; [24,28]), and rates of 97.6% in children (aged 6–14 years) with any stage of g-HAT (DNDiFEX006; [24,28]). However, among stage 2 patients with baseline WBCs in the CSF of >100 /µL, the treatment success rate at 18 months was found to be lower with fexinidazole than with NECT (86.9% versus 98.7%, respectively) [24,28]. A further clinical trial (DNDiFEX09HAT) is currently ongoing to assess the efficacy and safety of fexinidazole in population groups not included in the previous trials (including pregnant or breastfeeding women, and patients with poor nutritional status or with chronic diseases) and include both patients treated in health centers and those treated in an outpatient setting [18,27,37].

Fexinidazole treatment was well tolerated, particularly by patients with stage 1 disease, and pooled analysis of the safety data from the three clinical studies indicated that fexinidazole had an acceptable safety profile: the most common adverse events (AEs) considered as being related to fexinidazole were mild or moderate (vomiting, nausea, asthenia, decreased appetite, headache, insomnia, tremor, and dizziness) and only four serious AEs considered as possibly related to fexinidazole were reported (two reports of personality change, one of acute psychosis and one of hyponatremia) [24,27,28]. The pivotal clinical trial showed that there were no differences in the frequency of patients experiencing at least one AE between patients that received fexinidazole and those that received NECT (94% versus 93%, respectively) [39]. However, several of the most frequently observed treatment-emergent AEs occurred more commonly in the fexinidazole group than in the NECT group, including headache, insomnia, nausea, asthenia, tremor, and dizziness. Conversely, a lower proportion of patients taking fexinidazole than NECT experienced hyperkalemia, pyrexia, convulsions, and chills. Vomiting was reported in a similar percentage of patients in the fexinidazole (28.4%) and NECT groups (28.5%) [39]. With the exception of vomiting shortly after fexinidazole administration (20% of pediatric patients versus 6% of adult patients vomited within 30 min of drug intake), the safety profile of fexinidazole in the pediatric study population was similar that in adults [24].

The pivotal clinical trial was conducted at 10 sites in the DRC and Central African Republic (CAR) by African investigators under the guidance of the principal investigator and director of the NSSCP in the DRC, Dr V Kande ([39]; Figure 4). Extensive collaboration with partners of the HAT Platform, with support from the Swiss TPH and the Société Française et Francophone d’Éthique Médicale (SFFEM) were required to enable these clinical trials to be conducted in accordance with Good Clinical Practice guidelines and international ethics standards [38]. This was achieved by strengthening the capacity and improving the infrastructure of local facilities in the often remote geographical regions of the DRC and CAR (Figure 4), resulting in the renovation and refurbishment of nine referral treatment units, training of over 200 researchers, monitors and practitioners (Table 3), and provision of support and training to active surveillance teams for the screening of over 2 million people in the DRC [40].

### 3.3. A Positive Opinion from the European Medicines Agency

The dossier was submitted to the EMA on the 14 December 2017, under Article 58 of (EC) No. Regulation 726/2004 which allows medicines intended for markets outside the European Union to be evaluated by the EMA in the context of collaboration with the WHO and other non-EU authorities, such as regulators from the endemic African nations [38,41]. On the 15 November 2018, fexinidazole gained a positive scientific opinion from the EMA as an oral monotherapy for the treatment of both stages of g-HAT at the same dosing regimen [27,38,41]. This positive opinion paves the way for applications for marketing authorization and registration of fexinidazole in endemic countries, and for its free distribution by the WHO under the global alliance agreement made with Sanofi covering all anti-HAT medicines [9,27,42].

The involvement of the WHO and regulators from the DRC and Uganda in the EMA submission process under article 58 has facilitated the registration process in these countries: fexinidazole was approved for use in the DRC in December 2019, an unprecedented time-scale [42]. The registration process in Uganda is also underway.

The positive scientific opinion from the EMA was followed by the inclusion of fexinidazole in the WHO List of Prequalified Medicinal Products by March 2019 and then in the WHO Model List of Essential Medicines that same year [43]. The WHO has also updated its guidelines for the treatment of g-HAT [5], recommending that the 10-day oral treatment with fexinidazole is used as the first-line treatment for g-HAT patients aged ≥ 6 years (weighing ≥ 20 kg) with the exception of patients with a WBC > 100/µL in the CSF after lumbar puncture performed due to clinical suspicion of severe stage 2 disease. These updated guidelines specified the need for medically supervised administration and monitoring to ensure adequate food intake. It was also noted that convenience for the patient and their family, development of side effects, existing comorbidities and the capacity of the healthcare system needed to be considered in the decision to administer fexinidazole on an outpatient basis. Hospitalization was deemed mandatory for patients with psychiatric disorders or a history of psychiatric disorders, those at risk of poor compliance to treatment, children under 35 kg, and patients with a WBC > 100/µL in the CSF exceptionally treated with fexinidazole [5].

Animal toxicity data from the regulatory preclinical studies indicated that fexinidazole did not have any direct or indirect harmful effects on reproductive function or teratogenicity at therapeutic doses [24]. As a precautionary measure, the use of fexinidazole should be avoided during the first trimester of pregnancy, and fexinidazole should be used during the second and third trimesters only if the potential benefits outweigh the potential risks to both the mother and the fetus [24]. As pharmacokinetic studies indicate that fexinidazole and its metabolites are excreted in breast milk, the benefits of therapy for the mother and the benefits of breastfeeding for the child need to be considered before initiating fexinidazole treatment [24]. A clinical trial is currently ongoing to further assess the efficacy and safety of fexinidazole in patients with g-HAT, including pregnant and breastfeeding women.

### 3.4. The Cost of Developing a New Chemical Entity

The DND*i* estimates the cost involved in the development of fexinidazole, from data mining activities in 2005 through to submission of the EMA dossier in 2018–2019, at 55 million Euros (Figure 5a) [40]. Donations from a range of partners also played a key role in funding this project, with around 31 million Euros being donated by the Bill & Melinda Gates Foundation, and around 25 million Euros being donated by the UK Department for International Development, other public donors, MSF and other private donors (Figure 5b). In addition, Sanofi estimates a further 13 million Euros has been spent on regulatory, human resource and industrial activities during the course of fexinidazole development. However, the value of many of the contributions involved in the development of this new treatment cannot be included in these figures, such as free access to Sanofi’s assets and active pharmaceutical agents for trials, or the expertise and materials for research and development studies and product registration. The scale of the costs involved in the development of fexinidazole underscores the commitment undertaken by all the public and private partners that have supported the HAT elimination initiative, either through core funding or resources specifically earmarked for fexinidazole (Figure 5b), and highlights the true value of global collaboration to achieve this common goal.

## 4. Conclusions

The resounding successes of the HAT elimination initiative and the fexinidazole development project are undoubtedly due to the strategy of global alliance and collaboration with highly dedicated and motivated individuals and organizations, and public and private collaborators. In forming an alliance to combat 21st century HAT epidemics, the HAT program and fexinidazole development project have demonstrated the effectiveness of public–private partnership in tackling global health issues. Furthermore, NSSCPs played a major role in both the control and surveillance program, and during the clinical phase of fexinidazole development, leaving behind a legacy of improved health and research infrastructure. Following on from the success of the fexinidazole program, similarly collaborative frameworks–involving private–public partnerships—are ongoing to further improve therapeutic options for g-HAT (for example, acoziborole [44]).

## Figures and Tables

**Figure 1 tropicalmed-05-00017-f001:**
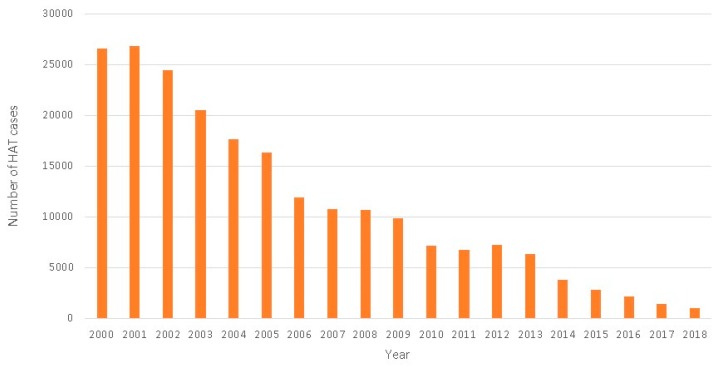
Total number of Human African Trypanosomiasis (HAT) cases reported per year (2000–2018) using data provided by national sleeping sickness control programs, non-governmental organizations, and research institutions, and assembled in the Atlas of HAT published in [2,15].

**Figure 2 tropicalmed-05-00017-f002:**
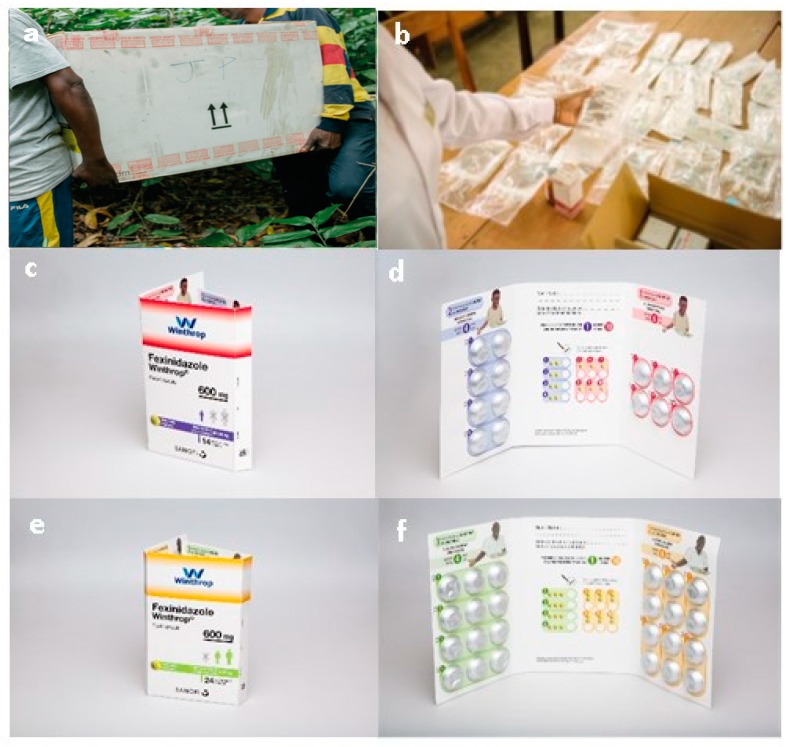
Latest treatment options for Human African Trypanosomiasis (HAT). (**a**) Heath workers transporting a Nifurtimox–Eflornithine Combination Therapy (NECT) kit in a remote region of the Democratic Republic of the Congo. The package contains four treatments and weighs approximately 36 kg [23]. (**b**) NECT required for the treatment of one patient. Nifurtimox is supplied in glass bottles, each containing 100 tablets of 120 mg and is given orally at a dose of 15 mg/kg/day, every 8 h for 10 days. Eflornithine is supplied in glass bottles (200 mg/mL in 100 mL bottles), is given by intravenous infusion 400 mg/kg/day every 12 h for 7 days, and once opened must be stored in the fridge (for up to 24 h) [23]. (**c**) and (**d**) Fexinidazole treatment for children (aged ≥ 6 years and weighing ≥ 20 and <35 kg): each wallet contains 14 tablets (1 aluminum foil blister of 6 tablets and 1 aluminum foil blister of 8 tablets) to be taken once daily with food as 2 tablets (1200 mg) once a day for the first 4 days of treatment, and 1 tablet (600 mg) once a day for the remaining 6 days. (**e**) and (**f**) Fexinidazole treatment for adults (weighing ≥ 35 kg): each wallet contains 24 tablets (2 aluminum foil blisters of 12 tablets) to be taken once daily with food with 3 tablets (1800 mg) once a day for the first 4 days of treatment, and 2 tablets (1200 mg) once a day for the remaining 6 days [24].

**Figure 3 tropicalmed-05-00017-f003:**
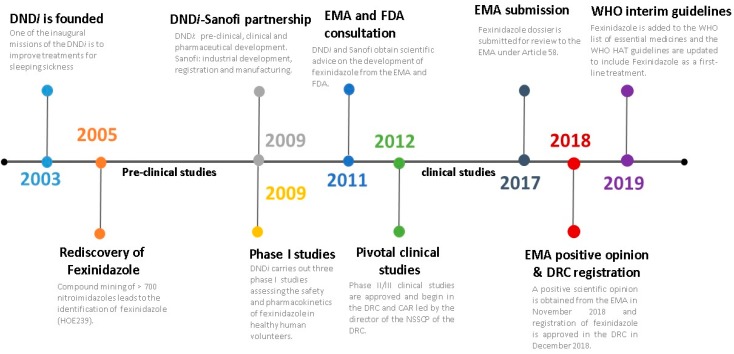
Timeline of the fexinidazole development project. DND*i*, Drugs for Neglected Diseases *initiative*; EMA, European Medicines Agency; FDA, Food and Drug Administration; DRC, Democratic Republic of Congo; CAR, Central African Republic; NSSCP, National Sleeping Sickness Control Program; WHO, World Health Organization; HAT, Human African Trypanosomiasis.

**Figure 4 tropicalmed-05-00017-f004:**
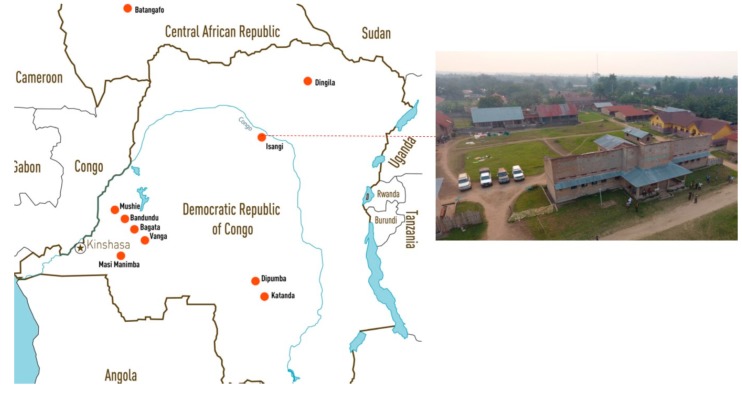
Locations of the 10 sites involved in the phase II/III clinical trials of fexinidazole led by the director of Le Programme National de Lutte contre la Trypanosomiase Humaine Africaine (PNLTHA). Inset: the Isangi clinical site along the Congo river in Democratic Republic of Congo.

**Figure 5 tropicalmed-05-00017-f005:**
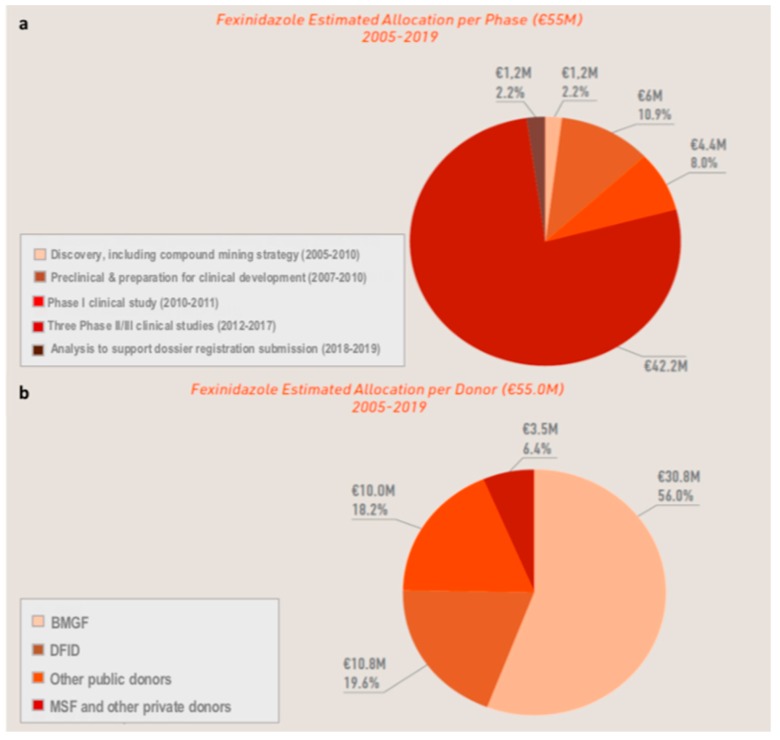
Estimated cost and funding of the fexinidazole development project. (**a**) Estimated allocation of funds per phase of development: 55 million Euros from 2005–2019. (**b**) Estimated allocation per donor. Donors: the Bill & Melinda Gates Foundation (BMGF, Seattle, WA, USA), the UK Department for International Development (DFID, London, UK), the Dutch Ministry of Foreign Affairs, the German Federal Ministry of Education and Research, the German Development Agency, the French Development Agency, the French Ministry of Foreign and European Affairs, the Norwegian Agency for Development Cooperation, the Spanish Agency for International Development Cooperation, the Republic and Canton of Geneva, Switzerland, the Swiss Agency for Development and Cooperation, Médecins Sans Frontières (MSF)/Doctors without Borders, Brian Mercer Charitable Trust, and other donors from the HAT campaign.

**Table 1 tropicalmed-05-00017-t001:** Summary of former treatments available for HAT before development of fexinidazole.

HAT Stage	Drug Name (Marketing Date)	Route, Dose	Comments
Stage 1	Pentamidine (1940)	Single IM dose of 4 mg/kg/day for 7 days (preferred) or IV injections	Skilled health workers, adverse events (metabolic disorders, pancreatitis, local abscesses), heavy treatment burden (for patients and families), parasite resistance. Effective in stage-1 g-HAT only
	Suramin (1920s)	Test dose then weekly IV injections for 5 weeks	Primarily for stage-1 r-HAT rarely used for g-HAT, adverse events (anaphylactic reactions, renal toxicity), heavy treatment burden (for patients and families)
Stage 2	NECT (2009)	Nifurtimox: 5 mg/kg PO every 8 h (15 mg/kg/day) for 10 days. Eflornithine: 200 mg/kg/day IV infusion every 12 h (400 mg/kg/day) for 7 days	Systematic hospitalization, skilled health workers, logistical concerns. First-line treatment for stage 2 g-HAT
	Eflornithine or DFMO (1990)	100 mg/kg/day IV infusion every 6 h (400 mg/kg/day) for 14 days	Systematic hospitalization, skilled health workers, long therapy, heavy treatment burden (for patients and families). Mainly used as a second-line treatment for stage-2 g-HAT
	Melarsoprol (1949)	2.2 mg/kg/day slow IV injection for 10 days	Highly toxic, painful, parasite resistance, skilled health workers. Restricted to cases refractory to NECT for stage-2 g-HAT. Remains the only drug effective in stage-2 r-HAT

Abbreviations: g-HAT, Human African Trypanosomiasis due to *T. b. gambiense*; IM, Intramuscular; IV, Intravenous; NECT, Nifurtimox-Eflornithine Combination Therapy; PO, Per Oral; r-HAT, Human African Trypanosomiasis due to *T. b. rhodesiense*.

**Table 2 tropicalmed-05-00017-t002:** Summary of phase II/III clinical trials evaluating the efficacy of fexinidazole in Human African Trypanosomiasis caused by *Trypanosoma brucei gambiense*.

Study Name, Designation, Phase and Duration	Study Description	Patients	Key Results
**DNDiFEX004 NCT01685827 (II/III: 2012–2015)**	Efficacy and safety of fexinidazole compared to NECT in patients (aged ≥ 15) with late stage 2 g-HAT: a pivotal, non-inferiority, multicenter, randomized, open-label study	394 (264 fexinidazole; 130 NECT)	Study confirms the efficacy of fexinidazole in late stage 2 g-HAT patients: success rate at 18 months post-treatment of 91.2% for fexinidazole, versus 97.6% for NECT within the margin of acceptable difference (97.06% CI −11.2 to −1.6; *p* = 0.0029). [18,39]
**DNDiFEX005 NCT02169557 (II/III: 2014–2017)**	Efficacy and safety of fexinidazole in patients (aged ≥ 15) with stage 1 or early stage 2 g-HAT: a prospective, multicenter, open-label and single arm cohort study, plug-in to the pivotal study	230	Studies confirm the efficacy of fexinidazole in stage-1 and early-stage-2 g-HAT patients: success rates in adults of 98.7% (95% CI [96.2%–99.7%]) and in children of 97.6% (95% CI [93.1%–99.5%] at 12 months post-treatment. [24]
**DNDiFEX006 NCT02184689 (II/III: 2014–2017)**	Efficacy and safety of fexinidazole in children (≥6 years and <15; ≥20 kg) with g-HAT of any stage: a prospective, multicenter, open-label and single arm cohort study, plug-in to the pivotal study	125	
**DNDiFEX09HAT NCT03025789\** **(IIIb:2016–ongoing)**	An open-label study assessing effectiveness, safety and compliance with fexinidazole in patients with g-HAT at any stage	174	Follow up ongoing [37]

Abbreviations: g-HAT, Human African Trypanosomiasis caused by *Trypanosoma brucei gambiense*; NECT, Nifurtimox–Efornithine Combination Therapy; CI, Confidence Interval.

**Table 3 tropicalmed-05-00017-t003:** Strengthening the local capacity of clinical sites in remote regions of the Democratic Republic of Congo and the Central African Republic.

**Infrastructure**	Renovation and refurbishment of 9 referral units in rural district hospitals Provision of solar energy equipment and generators Installation of Internet and satellite connections to transmit care report forms Improvement of waste management
**Supply of Equipment**	Medical equipment: oxygen concentrators, defibrillators, ECG devices etc. Diagnostic equipment: microscopes, Piccolo analyzers etc. Incinerators for waste management
**Training of Staff**	Good Clinical Practice guidelines Universal standard precautions Laboratory diagnosis Patient examination techniques and treatment procedures Pharmacovigilance Waste management

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
