# Peer review of "Innovative Partnerships for the Elimination of Human African Trypanosomiasis and the Development of Fexinidazole"

_tropicalmed, 2020, doi:10.3390/tropicalmed5010017_

Round 1

Reviewer 1 Report

In this review the Authors describe the path that led to the approval of Fexinidazole, the first all-oral drug capable of curing both stages of the neglected disease Human African Trypanosomiasis. The special focus on the efforts of the international partners involved in this feat and the long-term benefits that the drug development process brought to the endemic countries where trials have been carried out, make this review relevant. Despite the need for some spell checks and possibly the addition of some references, the paper is well written and documented and I recommend it for publication after the points listed below have been addressed.

The Authors mostly cite websites, press releases and reports by some of the major actors involved (as WHO, DNDi and EMA). Some extra relevant publications could be added to the manuscript though, such as:

-Deeks, E.D. Drugs (2019) 79: 215. https://doi.org/10.1007/s40265-019-1051-6.

-The European Medicines Agency's scientific opinion on oral fexinidazole for human African trypanosomiasis. Pelfrene E, Harvey Allchurch M, Ntamabyaliro N, Nambasa V, Ventura FV, Nagercoil N, Cavaleri M. PLoS Negl Trop Dis. 2019 Jun 27;13(6):e0007381. doi: 10.1371/journal.pntd.0007381. eCollection 2019 Jun.PMID:31246956.

-Chappuis F. Oral fexinidazole for human African trypanosomiasis. Lancet, 2018 Jan 13;391(10116):100-102. doi: 10.1016/S0140-6736(18)30019-9.

The manuscript presents with many small typos. Please check and correct the following:

-Line 27: spell the acronym DNDi (as this is its first mention in the manuscript)

-Line 62: add “s” to 1960

-Line 64: change “but that” with “although”

-Lines 83-84: substitute “;” with a comma after “WHO” and after “treatment”

Line 100: only g-HAT has been targeted for elimination, not r-HAT. Specify this by changing “HAT” to “g-HAT” (same in Line 113)

Line 126: add “the” before “presence”

Line 126: eliminate “in the” (repetition)

Please correct the Font size of the following:

-Lines 40-42: T. b. gambiense and T. b. rhodesiense

-Line 182: H. Hänel (also not italics)

-Line 183: Bourdin Trunz (also not italics)

-Line 184: Hänel (also not italics)

-Line 185: Hoechst AG Frankfurt/Main (also not italics)

-Line 186: Hoechst (also not italics)

Lines 190-192: eliminate italics formatting

Line 195 (Fig 3): DRC Republic with capital letter

Line 216: add space between > and 90%

Line 246: Ethique should be Éthique

Table 1:

-substitute dot with comma after anaphylactic reactions (Suramin section)

-eliminate “and second stage” after “Stage 2” (on the left of the table)

Figure 2:

-Move a) before NECT. Add a full stop after ref 20 (line142). Move c) before “fexinidazole” (line 146) and e) and f) before fexinidazole in line 149

The formatting of the references should be re-checked. Some corrections follow here:

-Line 352 (ref 2), Line 378 (ref 12), Line 381 (ref 13), Line 394 (ref 17), Line 431 (ref 31): PLoS neglected tropical diseases should be PLoS Negl Trop Dis

-Line 358 (ref 5), check and correct Formatting (WHO added at the beginning. The same should be done for all other similar references such as ref 8, 9, etc)

-Line 364 (ref 7), the correct reference should be “WHO Expert Committee on the Control and Surveillance of African Trypanosomiasis & World Health Organization. Control and surveillance of African trypanosomiasis: report of a WHO expert committee. World Health Organization, (‎1998)‎. The technical report series details should be added too.

-Line 390: eliminate one dot after Committee (“Commitee” need an extra “T” too)

-Line 400 (ref 19): Pathogens and global health should be Pathog Glob Health

-Line 406 (ref 22): add dot after sleeping sickness

-Line 409 (ref 23): Trends in parasitology should be Trends Parasitol

-Line 416 (ref 25): end of sentence missing

-Line 417-418: Parasitology research should be Parasitol Res

-Line 423: T. brucei should be in italics

-Ref 30: species names in italics and journal name abbreviated

-Line 441: Trypanosoma brucei gambiense in italics

Other corrections/small changes required are listed below:

Line 131: please add a reference for the introduction of NECT. Also specify its use for g-HAT, not HAT in line 133.

Lines 161-162: the doses for children indicated are incorrect: the sentence should be corrected to “1200 mg/d loading dose for 4 days and 600 mg/d maintenance dose for 6 days”.

Lines 167-173: please add and discuss the WHO guideline observation concerning the need for monitoring of drug intake with food (i.e.- from WHO guidelines for g-HAT “Directly observed treatment: Each intake of fexinidazole must be supervised by a trained health staff who must ensure that the patient is in a fed condition. Outpatient administration (under daily supervision) can be decided in consultation with the patient, his/her family and clinicians, taking into account the following factors: convenience to the patient and the family (e.g. distance and costs); development of side-effects interfering with treatment compliance; existing comorbidities; and capacity of the healthcare system for supervised administration as an outpatient. Hospitalization should be mandatory in the following cases: patients with psychiatric disorders; children with body weight < 35 kg; patients with ≥ 100 WBC treated (exceptionally) with fexinidazole; and risk of poor compliance with treatment.”).

Reviewer 2 Report

This is an interesting and timely piece concerning the new therapy for HAT. I learned a lot about the background, and while a historic piece, is a valuable record. I have only minor suggestions as the work is well written, lively and a pleasure to read. 

Line17 As we are discussing the drop of impact of trypanosomiasis, perhaps 'was' rather than 'is'

I felt that the mention of the other new HAT drug, acosiborole, would be appropriate - very brief but ensures the timeliness of this work. 

Disussion of the spend from Sanofi on P2 is fine and justified indeed, but I did also feel that given the major contributions from Gates, MSF and others perhaps this should also be mentioned? Sort of "By comparison...."

P3 - within the need for new drugs a mention perhaps also of shelf life and emergence of resistance would be valid - this is a major issue with some of the canonical drugs and has been a growing concern. Also, do the authors have any information concerning the likelihood of resistance of fexinidazole? 

L187 delete 'the'

Unclear what an acceptability margin would be and what 13% represents. 

Any information on the treatment failures discussed on line 222 and following? 

L256 Define EMA

Line 282 such AS free

Reviewer 3 Report

> The review article is very interesting and shows how Fexinidaozle
> reached the therapeutic application with the collaboration of many
> institutions and companies during the years fo its development.
> Two main questions remain open:
> 1. a better description of the potential effect on pregnant women
> (teratogenicity) and consequent safety concerns
> 2. safety concerns more deeply discussed in comparison with NECT
> 3. pediatric use of the drug: how was it developed and wich concerns.
>
> Some repetitions are present along with the manuscript.
>
> These three points are essential to complete the drug profile
> presentation and before declaring that Fexinidazole is completely
> satisfying the medical need.
>

Round 2

Reviewer 3 Report

There is no doubt about the improvement and wonderful collaborative efforts performed.

Therefore I find all the story an excellent example of neglected tropical drug development for low-income countries. However, I think that the message about the compound profile should be direct and transparent to simplify the application.

My non-binding opinion (because it includes a comment about WHO release) is the following:

 325 " Due to a lack of data on the use of fexinidazole in pregnant and breastfeeding women [24], the updated guidelines include precautionary recommendations against administration of fexinidazole during the first trimester, and advise consideration of the risks and benefits before use of fexinidazole during the second and third trimesters"

This sentence is unclear in the concepts: Ethical consideration suggests that the best sentence would be the following one: Fexinidazole may show a high risk of teratogenicity and cannot be used in case of pregnancy. Same thing for breastfeeding mothers. This is usually written in drug indications.  I cannot see why in this case it cannot be used.

The term "Lack of data on the use ..." apparently suggests that the outcome of the treatment will provide information about drug toxicity on pregnant women: of course, this looks horrible for many reasons.   Moreover, it looks like an unwanted discriminatory attitude, and it is better to avoid.

Please change the sentence writing definitely what was written in the letter of response to my attention. That was very clear, just elaborate properly for the review.

One possibility:

The authors wrote in their letter to my attention as follows:

" Animal toxicity data do not indicate (direct or indirect) harmful effects with respect to reproductive function or teratogenicity at therapeutic doses. As a precautionary measure, the use of fexinidazole should be avoided during the 1st trimester of pregnancy (very clear), and fexinidazole should be used during pregnancy in the 2nd and 3rd trimesters only if the potential benefits to the mother outweigh the potential risks, including those to the fetus (very unclear this latest sentence). A further clinical trial is currently ongoing to assess the efficacy and safety of fexinidazole in pregnant and breastfeeding women.(Important because justifies the first sentence)

I would avoid the inclusion of WHO recommendations because of the reasons highlighted above.

----

Another aspect: Psychiatric use administered under control to ensure adherence to therapy, like any other drug, therefore I cannot understand the specific recommendation highlights.

Author Response

We thank the reviewer for their positive comments on the revised version of our manuscript.

There is no doubt about the improvement and wonderful collaborative efforts performed.

Therefore I find all the story an excellent example of neglected tropical drug development for low-income countries. However, I think that the message about the compound profile should be direct and transparent to simplify the application.

My non-binding opinion (because it includes a comment about WHO release) is the following:325 " Due to a lack of data on the use of fexinidazole in pregnant and breastfeeding women [24], the updated guidelines include precautionary recommendations against administration of fexinidazole during the first trimester, and advise consideration of the risks and benefits before use of fexinidazole during the second and third trimesters"

This sentence is unclear in the concepts: Ethical consideration suggests that the best sentence would be the following one: Fexinidazole may show a high risk of teratogenicity and cannot be used in case of pregnancy. Same thing for breastfeeding mothers. This is usually written in drug indications.  I cannot see why in this case it cannot be used.

The term "Lack of data on the use ..." apparently suggests that the outcome of the treatment will provide information about drug toxicity on pregnant women: of course, this looks horrible for many reasons.   Moreover, it looks like an unwanted discriminatory attitude, and it is better to avoid.

Please change the sentence writing definitely what was written in the letter of response to my attention. That was very clear, just elaborate properly for the review.

One possibility:

The authors wrote in their letter to my attention as follows:

" Animal toxicity data do not indicate (direct or indirect) harmful effects with respect to reproductive function or teratogenicity at therapeutic doses. As a precautionary measure, the use of fexinidazole should be avoided during the 1st trimester of pregnancy (very clear), and fexinidazole should be used during pregnancy in the 2nd and 3rd trimesters only if the potential benefits to the mother outweigh the potential risks, including those to the fetus (very unclear this latest sentence). A further clinical trial is currently ongoing to assess the efficacy and safety of fexinidazole in pregnant and breastfeeding women.(Important because justifies the first sentence)

I would avoid the inclusion of WHO recommendations because of the reasons highlighted above.

The sentence has been changed to provide more precise details of the information provided in the Summary of Product Characteristics for Fexinidazole (https://www.ema.europa.eu/en/documents/medicine-outside-eu/fexinidazole-winthrop-product-information_en.pdf): Animal toxicity data from the regulatory preclinical studies indicated that fexinidazole did not have any direct or indirect harmful effects on reproductive function or teratogenicity at therapeutic doses [24]. As a precautionary measure, the use of fexinidazole should be avoided during the first trimester of pregnancy, and fexinidazole should be used during the second and third trimesters only if the potential benefits outweigh the potential risks to both the mother and the fetus [24]. As pharmacokinetic studies indicate that fexinidazole and its metabolites are excreted in breast milk, the benefits of therapy for the mother and the benefits of breastfeeding for the child need to be considered before initiating fexinidazole treatment [24]. A clinical trial is currently ongoing to further assess the efficacy and safety of fexinidazole in patients with g-HAT, including pregnant and breastfeeding women.

----

Another aspect: Psychiatric use administered under control to ensure adherence to therapy, like any other drug, therefore I cannot understand the specific recommendation highlights.

The revisions concerning supervised administration during outpatient administration and description of the recommendations for hospitalized treatment were made in response to comments from another reviewer.
